# Predicting Presence of Amphibian Species Using Features Obtained from GIS and Satellite Images

**Marcin Blachnik** [1,*] , **Marek Sołtysiak** [2] **and Dominika Dąbrowska** [2]

[1] Department of Industrial Informatics, Silesian University of Technology, Katowice 43-100, Poland
[2] Faculty of Earth Sciences, University of Silesia, Sosnowiec 41-200, Poland; marek.soltysiak@us.edu.pl (M.S.); dominika.dabrowska@us.edu.pl (D.D.)
[*] Correspondence: marcin.blachnik@polsl.pl; Tel.: +48-32-603-4172

**Abstract:** The construction of transport infrastructure is often preceded by an environmental impact assessment procedure, which should identify amphibian breeding sites and migration routes. However, the assessment is very difficult to conduct because of the large number of habitats spread out over a vast expanse, and the limited amount of time available for fieldwork. We propose utilizing local environmental variables that can be gathered remotely using only GIS systems and satellite images together with machine learning methods. In this article, we introduce six new and easily extractable types of environmental features. Most of the features we propose can be easily obtained from satellite imagery and spatial development plans. The proposed feature space was evaluated using four machine learning algorithms, namely: a C4.5 decision tree, AdaBoost, random forest and gradient-boosted trees. The obtained results indicated that the proposed feature space facilitated prediction and was comparable to other solutions. Moreover, three of the new proposed features are ranked most important; these are the three dominant properties of the surroundings of water reservoirs. One of the new features is the percentage access from the edges of the reservoir to open areas, but it affects only a few species. Furthermore, our research confirmed that the gradient-boosted trees were the best method for the analyzed dataset.

**Keywords:** amphibians; water reservoirs; GIS; machine learning

## 1. Introduction

Aquatic ecosystems occupy an important role in the natural environment. Water reservoirs, wetlands and river valleys are habitats for numerous valuable natural species of flora and fauna. Aquatic ecosystems affect the microclimate, delay the flow of surface water, supply groundwater and contribute to the improvement of water quality [1–3].

Sustainable development highlights the need to seek a compromise between economic development and environmental protection. Responsible space management requires the valorization of the environment to be carried out in order to indicate natural valuable areas, including aquatic ecosystems. Areas with low natural value can be developed as a potential investment. Amphibians are very sensitive bio-indicators of the environment [1], and among the phyla of vertebrates [2–7] they are most threatened by extinction. Amphibians are closely dependent on water reservoirs where they breed. They need good quality water and suitable habitat conditions [2–4]. Moreover, amphibians are considered to be valuable elements of the natural environment, as reflected in the second and fourth Annexes to the European Habitats Directive, which list 20 and 45 species, respectively.

There are 18 species of amphibians in Poland and only the fire salamander uses flowing waters (small streams with low currents) as a reproduction site. Amphibians develop attachments to water reservoirs—they return to their birthplace to breed. This phenomenon is called philopatry. During

the mating season, almost all adults of a given amphibian population gather in water reservoirs, thus an estimate of their numbers can be made. The breeding season for individual species falls within different periods of spring and may take place for a longer or shorter period, depending on the species. For instance, the common toad is a typical terrestrial species that spends only a few days in a water reservoir in order to reproduce. However, green frogs tend to stay in water reservoirs until autumn, given that their mating season lasts from the end of April to the end of June [2].

In addition, the land habitat, typically located near the water reservoir, serving as a breeding site for amphibians is of equal importance. The land habitat must provide them with an opportunity to feed, hide and survive winter. Amphibians are not very mobile and in general do not move further than 1500 m from their breeding sites [4]. It is difficult to estimate the number of amphibians on land when they are scattered. For this reason, an estimate of the number of amphibians can be made while they are in the water reservoirs. Hence, water reservoirs are important because they allow amphibian reproduction and, at the same time, they facilitate the examination of amphibian populations. Assuming that the value of a population depends on the condition of the habitat, when assessing a habitat, we can draw inferences about the value of the population. This assumption forms the basis for the later portion of the paper.

The current state of law requires that careful preparations for any investment be made. pursuant to the European Environmental Impact Assessment Directive. For investments that may have a significant impact on the environment an environmental impact assessment (EIA) is required. This includes, among others, the determination of the influence on protected species of fauna and their habitats.

Irrespective of the permit procedure for investments, it is necessary to recognize field conditions. In the case of amphibians, fieldwork is mainly carried out in the areas of reservoirs in which amphibians breed and develop. In order to gather representative data, it is necessary to use appropriate methodology. The actual richness of amphibian populations is quite difficult to determine [8]. Difficulties in determining the richness of amphibians may be due to [5]:

1. Difficult terrain conditions—extensive surfaces and overgrown reservoirs. In general, the large surface of the reservoirs means that the observation field in which amphibians may be found covers a relatively wide area. In the overgrown areas, some pygmies remain invisible to the observer, and thus may go undetected. The development of vegetation in water reservoirs limits access to the observation points. In the case of large reservoirs, the length of the shoreline increases and more time is needed to conduct the observations. With a large number of sites to investigate, this remains a very important factor, given that the breeding period of some amphibian species, like the common frog, moor frog, or common toad, is short. With a limited time period, it is possible to underestimate the number of specimens or even to omit the occurrence of certain species.

2. The secretive lifestyle of some species—this applies mainly to newts, which do not make mating noises, stay under the water surface and appear on the surface only for a short time to draw air. It is also difficult to estimate the common spadefoots, which produce quiet sounds from under the water.

3. Inconvenient and unusual weather conditions (e.g., cold and dry springs)—in a typical spring season, amphibian species start their breeding period in a specific order. However, in the case of weather anomalies with prolonged low temperatures, breeding migrations may be slowed down or even stopped, and then accelerated after a warming period occurs. As a consequence, the breeding season of species that breed in early spring may be very short. In the case of snowless winters or a lack of rainfall, some breeding sites may be dry in the spring, which will make it difficult to find amphibians.

4. Natural fluctuations in the numbers of amphibians (e.g., after a long, cold winter). In the case of cold and snow-free winters, the abundance of amphibians is frequently reduced. Periodic changes in the population size are typical of amphibians. Due to their high reproductive potential, they can rebuild the population size. However, in an unfavorable year, the value of the population may be incorrectly assessed. Thus, it would be advisable to carry out research for at least two seasons, however, in the case of inventories taken for planned investments, this is not practiced.

5. A large number of research points spread out over a vast expanse of territory. This applies in particular to the species residing briefly in their breeding sites; in Poland, this concerns the common frog, the moor frog and the grey toad. The urgency due to the extent of the inventory area may lead to the non-detection of amphibians at certain sites (see point 1, 4 and 6).

6. A limited amount of time available for fieldwork (a few months or one season) and an insufficient number of researchers engaged in the fieldwork. Due to cost-cutting efforts, these two factors are quite common.

It would be desirable to devise an inspection mechanism allowing for an initial verification of the analyzed area, for the occurrence of habitats and the facilitation and assessment of the probability of occurrence of amphibians.

This article provides an example of the application of four data driven models to determine the influence of local environmental variables on the evaluation accuracy of amphibian distribution near water reservoirs in the area of road projects. Most of the proposed features can be easily obtained from the satellite imagery (http://satelliteworldmap.com) and urban spatial development plans (e.g., http://www.czestochowa.pl/page/3215,miejscowe-plany-zagospodarowania-przestrzennego.html). The six proposed new types of features characterizing water reservoirs are critical for a precise assessment of the occurrence of amphibian species. These are: three dominant types of surroundings of water reservoirs, the percentage of access from the edges of the reservoir to open areas, the state of maintenance of the reservoir and the type of shore.

The development of a species distribution model using data driven methods would be a useful tool in spatial planning, carrying out a strategic environmental impact assessment (EIA) and preparing a report. The described method can be used in particular in the process of planning of road investments, as it facilitates the detection of particular amphibian species based on known local environmental variables. It can be a useful tool for state institutions verifying the correctness of the implementation of a natural inventory. Moreover, the advantage of the suggested method lies in performing some initial analyses based on publicly accessible satellite maps (e.g., https://www.geoportal.gov.pl/). Furthermore, this method may help identify valuable positions of amphibians outside the scope of a natural inventory.

This paper discusses important new aspects such as:

- The introduction of methodology which allows us to assess the presence or absence of amphibian species remotely using pre-trained machine learning systems and variables obtained from satellite maps and GIS systems
- The use of six new types of variables, which were indicated by herpetologists dealing with the EIA

The main goal of the article is to present six new features that can be used to assess the occurrence of amphibians and to indicate the best methods of machine learning to predict these data.

## 2. Current State of Knowledge

Species distribution modeling [9] has become a useful method for predicting amphibian ranges, based on the relationships between species records and environmental variables. However, the state-of-the-art solutions reported by many authors cover very large geographical areas, which are not precise enough to allow us to accurately predict the absence or presence of given amphibian species in a single water reservoir. To ensure precise predictions, the local properties of the reservoirs which significantly affect the occurrence of amphibians are needed. To date, these have been taken into account but only to a limited extent (distance to road and surface of reservoirs). The issue of species distribution modeling has been discussed by many authors, either by the covering of a large area of research thus ensuring high generalizability [10–15], by only involving a GIS framework with the creation of a map of species occurrence but without predictions [16–19], or by using the maximum entropy algorithm for making predictions [20–23]. The authors have used between two and 20 variables [13,23–25].

Variables used by scientists can be divided into the following:

I. Those related directly to the climate [20] referred to annual temperature ranges, isothermality (the mean diurnal range/annual temperature range), annual mean precipitation, precipitation of the warmest quarter, coefficient of variation of monthly precipitation, annual total radiation, annual radiation range, and coefficient of variation of monthly relative humidity.

II. Those related to land use and the features of reservoirs [19] referred to the presence/absence of fringing vegetation in waterbodies, the presence/absence of floating and emergent vegetation in waterbodies, the waterbody size, the number of waterbodies within a 750 m radius, waterbody type and waterbody origin.

III. Finally, in the case of [26], those related to water state referred to hydroperiod, maximum water depth, water physicochemistry ($Cl^-$, $SO_4^{2-}$, $Na^+$, $K^+$, $Mg^{2+}$, $Ca^{2+}$), planktonic chlorophyll a, dissolved inorganic phosphate, nitrogen compounds, site characteristics and the potential connectivity of a pond.

The most frequently used variables are those related to land use—Type II [16,22,27–32]—due to the fact that they can be determined based on digital maps using GIS methods.

Some of the variables proposed by other authors, which belong to Type I features, such as temperature, landform or rainfall over a small area, cannot be applied to our problem. This is due to the fact that the values of these variables do not show high variability. The selected variables also require the reading of measurements from field-mounted sensors, which in turn requires financial outlays. Moreover, there are variables related to the characteristics of water in reservoirs, like physicochemistry properties (Type III variables), which can only be checked via biological or chemical tests [26]. Bearing in mind the variables proposed by other authors and the potential to verify the nature inventory, we decided to select only those variables that could be read directly from satellite maps. It should also be pointed out that the selected variables are not the only ones that can significantly affect the occurrence of amphibians.

In this article, several of the above-mentioned variables are used (see Section 3.1.2), and they are supplemented with six new ones, which can be extracted from the available maps and which, in the opinion of the experts, are important for the assessment of the natural habitats of amphibians.

## 3. Material and Methods

In the previous section, various concepts of spatial distribution modelling were briefly discussed. In our approach we focused on predictive modelling, using machine learning methods to utilize data that could be defined by an interpretation of satellite images. The general concept is depicted in Figure 1.

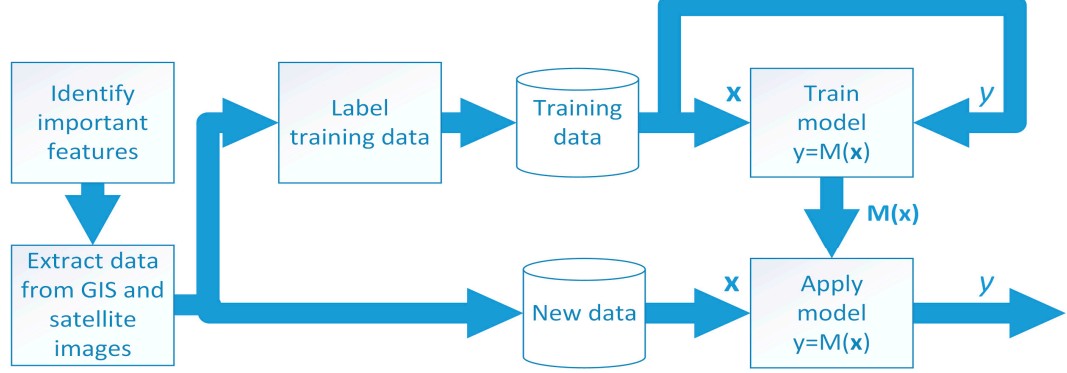

**Figure 1.** A scheme of the concept of machine learning algorithms applied to species distribution modelling.

The process starts by selecting appropriate features which indicate and affect the presence of various amphibian species near water reservoirs. Then, these features were used to describe the amphibian sites, so that each habitat constituted a single element in the training set labelled with the

presence/absence of a given species. Next, this dataset was used to create a set of predictive models, one per species. These models were then used to assess new habitats, described by the same set of features indicating the presence of each of the amphibians.

Below we provide information on the source of the data, the type of features used to describe each site and the machine learning methods and measures used in the experiments.

### 3.1. The Dataset

The data used in the experiments was derived from GIS and satellite information, as well as from information gathered from the natural inventories that were prepared for the EIA reports for two planned road projects (Road A and Road B) in Poland [33–37]. These reports were mostly used to gather information on the size of the amphibian population in each of the 189 occurrence sites.

### 3.1.1. The Study Area

The first road project concerned part of the planned A1 motorway section in Pyrzowice; the section is located along the northern border of the Silesian Voivodship and is about 75 km long (Figure 2). The field research involved a strip of land with a width of 500 m on both sides of the proposed project area. The field inventory was carried out in 2010 [33] and 2011 [34,35]. The results of these inventories were complemented by our own observations, which were conducted between 2014 and 2016 [36]. Finally, the first project included 81 amphibian breeding sites.

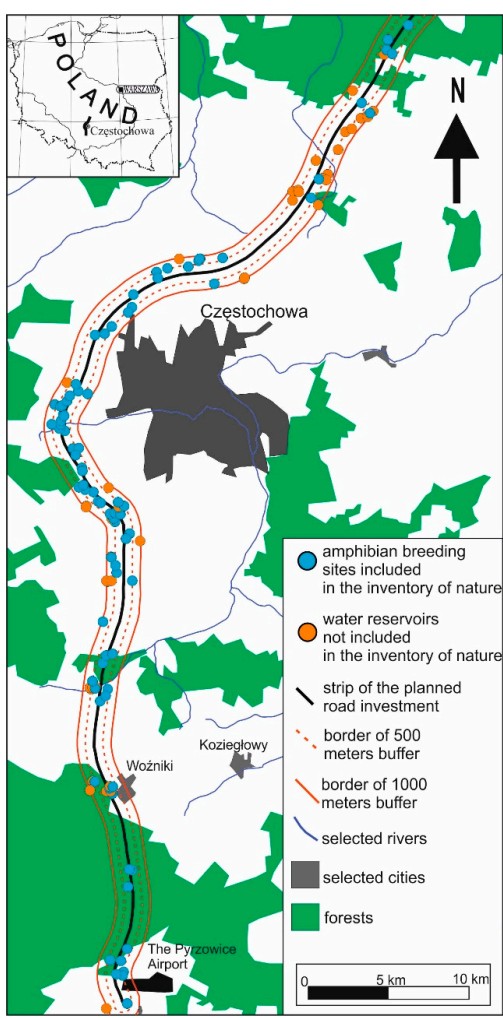

**Figure 2.** Localization of the section of the A1 motorway under analysis.

The second inventory was prepared in the vicinity of two variants of the planned Beskidy Integration Way on the Bielsko Biała-Wadowice-Głogoczów section of the S52 motorway. The length of this section of road is approximately 60 km. During the inventory, which was taken in 2010, 125 real and potential amphibian occurrence sites were described [37]. The methodology of the herpetological inventory included map analysis, literature and archive data analysis, and, then, field observations. As in the first case, the inventory was made in the spring time and consisted of the observation of the occurrence of amphibians in water reservoirs. The research area included a 500 m wide belt for each of the considered variants of the planned road. In order to conduct the final experiments, 108 amphibian occurrence sites were taken into account (Figure 3).

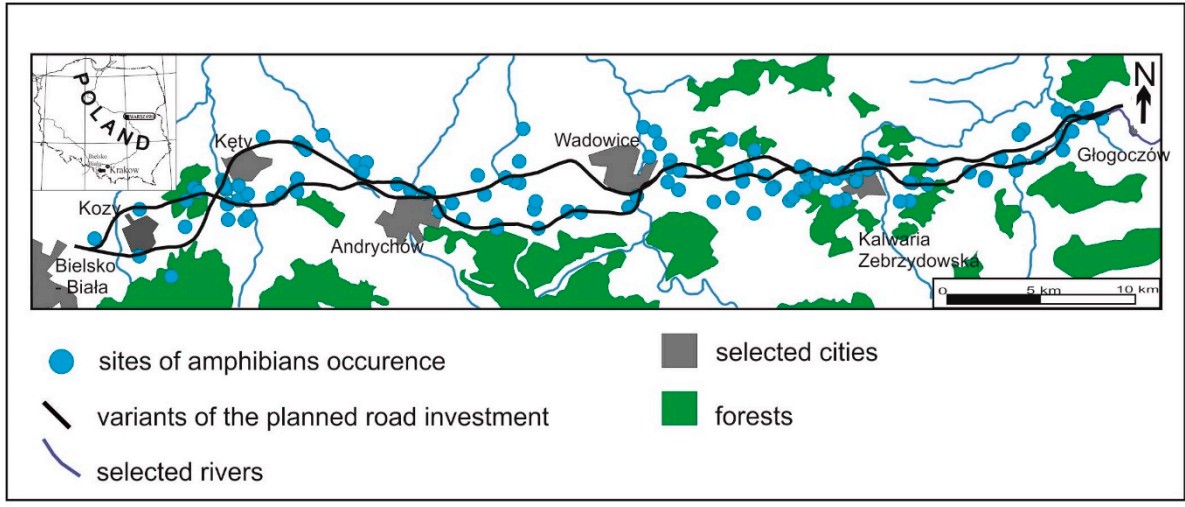

**Figure 3.** Localization of amphibian occurrence sites along planned s52 expressway.

### 3.1.2. Structure of the Dataset

The dataset was created based on herpetologists' experience of the assessment of amphibious habitats. They selected a set of features that covers the most aspects which affect the occurrence of amphibians in a certain area. These were:

1.  Surface of water reservoirs
2.  Number of water reservoirs in the habitat
3.  Type of water reservoirs
4.  First/second/third most dominant type of land cover in water reservoir surroundings
5.  Type of shore
6.  Intensity of vegetation in water reservoirs
7.  Maintenance status of the reservoir
8.  Use of water reservoirs
9.  Presence of fishing (using a high-resolution map)
10. Distance to buildings
11. Distance to roads
12. Percentage access from the edges of the reservoir to open areas

Features 1, 2, 10 and 11 are numerical (N) or ordinal (O), while the rest are categorical (C). Detailed descriptions of the features and the values they take are presented in Table 1.

**Table 1.** Features and its properties used to assess amphibian habitats.

| ID | Name | Symbol | Type | Description |
|---|---|---|---|---|
| 1 | Surface of water reservoir | SR | N | Surface of the water reservoir [$m^2$] |
| 2 | Number of water reservoirs in habitat | NR | N | More than 1 water reservoir is often situated in many habitats. Comment: The larger the number of reservoirs, the more likely it is that some of them will be suitable for amphibian breeding. |
| 3 | Type | TR | C | Type of water reservoirs: (a) reservoirs with natural features that are natural or anthropogenic water reservoirs (e.g., subsidence or post-exploited water reservoirs), not subjected to naturalization (b) recently formed reservoirs, not subjected to naturalization (c) settling ponds (d) water reservoirs located near houses (e) technological water reservoirs (f) water reservoirs in allotment gardens (g) trenches (h) wet meadows, flood plains, marshes (i) river valleys (j) streams and very small water courses Comment: The most valuable reservoirs for amphibians are reservoirs with the least anthropopressure. |
| 4 | Surroundings 1—the dominant types of land cover surrounding the water reservoir | SUR1 | C | The "surroundings" feature was designated to three stages. First, the dominant surroundings were selected. Then, two secondary types were chosen. (a) forest areas (with meadows) and densely wooded areas (b) areas of wasteland and meadows |
| 5 | Surroundings 2—the second most dominant types of land cover surrounding the water reservoir | SUR2 | C | (c) allotment gardens (d) parks and green areas (e) dense building development, industrial areas (f) dispersed habitation, orchards, gardens (g) river valleys |
| 6 | Surroundings 3—the third most dominant types of land cover surrounding the water reservoir | SUR3 | C | (h) roads, streets (i) agricultural land Comment: The most valuable surroundings of water reservoirs for amphibians are areas with the least anthropopressure and proper moisture. |
| 7 | Type of shore | CR | C | Natural or concrete Comment: A concrete shore of a reservoir is not attractive for amphibians. A vertical concrete shore is usually a barrier for amphibians when they try to leave the water. |
| 8 | Intensity of vegetation development | VR | C | Presence of vegetation within the reservoirs: (a) no vegetation (b) narrow patches at the edges (c) areas heavily overgrown (d) lush vegetation within the reservoir with some part devoid of vegetation (e) reservoirs completely overgrown with a disappearing water table Comment: The vegetation in the reservoir favors amphibians, facilitates breeding and allows the larvae to feed and give shelter. However, excess vegetation can lead to the overgrowth of the pond and water shortages. |
| 9 | Maintenance | MR | C | Maintenance status of the reservoir: (a) clean (b) slightly littered (c) reservoirs heavily or very heavily littered Comment: Trash causes devastation of the reservoir ecosystem. Backfilling and leveling of water reservoirs with ground and debris should also be considered. |

**Table 1.** *Cont.*

| ID | Name | Symbol | Type | Description |
|----|------|--------|------|-------------|
| 10 | Management of the water reservoir by man | UR | C | Use of water reservoirs:<br><br>(a)  unused by man (very attractive for amphibians)<br>(b)  recreational and scenic (care work is performed)<br>(c)  used economically (often fish farming)<br>(d)   technological |
| 11 | Fishing | FR | C | The presence of fishing:<br><br>(a)  lack of or occasional fishing<br>(b)  intense fishing<br>(c)  breeding reservoirs<br><br>Comment: The presence of a large amount of fishing, in particular predatory and intense fishing, is not conducive to the presence of amphibians. |
| 12 | Building development | BR | N/0 | Minimum distance to buildings:<br><br>1.  <50 m<br>2.  50–100 m<br>3.  100–200 m<br>4.  200–500 m<br>5.  500–1000 m<br>6.  >1000 m<br><br>Comment: The more distant the buildings, the more favorable the conditions for the occurrence of amphibians. |
| 13 | Roads | RR | N/0 | Minimum distance from water reservoir to roads:<br><br>1.  <50 m<br>2.  50–100 m<br>3.  100–200 m<br>4.  200–500 m<br>5.  500–1000 m<br>6.  >1000 m<br><br>Comment: The greater the distance between the reservoir and the road, the more safety for amphibians. |
| 14 | Access from water table to land habitats | OR | C | Percentage access from the edges of the reservoir to undeveloped areas (the proposed percentage ranges are a numerical reflection of the phrases: lack of access, low access, medium access, large access to free space):<br><br>(a)  0–25%—lack of access or poor access<br>(b)  25–50%—low access<br>(c)  50–75%—medium access,<br>(d)  75–100%—large access to terrestrial habitats of the shoreline is in contact with the terrestrial habitat of amphibians. |

The selected features define the degree of attractiveness of the habitat for amphibians. The basic condition for their occurrence is the presence of a water reservoir in the vicinity of attractive habitat land [38]. The most attractive reservoirs for amphibians are those with natural features and those not used by humans or not used in an extensive manner. The presence of vegetation in a water reservoir reinforces the occurrence of amphibians. Similarly, free access to the reservoir from the land habitat is very desirable. Wastelands, wetlands and wet forest habitats are attractive as well. A negative impact on amphibians will be caused by the presence of a busy road, a high density of buildings in the vicinity of the reservoir, or intensive fishing. These features can be determined by a satellite map analysis and during a single inspection of the site. Geodetic services and GIS applications have tools to determine parameters such as surface or distance. It is a valuable practice to use several sources of information—the satellite maps presented may have come from different periods and show the variability of the analyzed features of the habitat. Field inspection is very important for proper assessment of the habitat features. Satellite images do not contain all information, such as the presence of fish, fishing activity, or rubbish, and, in addition, they may be out of date.

The assessment of a sample water reservoir using the proposed variables is presented in Figure 4. The source documentation that was used to build the dataset contained a number of individual in

population habitats. However, these values were very imprecise, so we replaced them with a binary indicator determining presence or absence of given amphibian species.

The documentation mentions 11 species. These are the common frog *Rana temporaria,* the edible frog *Pelophylax kl. esculentus*, the pool frog *Pelophylax lessonae*, the marsh frog *Pelophylax ridibundus*, the common toad *Bufo bufo,* the green toad *Bufo viridis*, the fire-bellied toad *Bombina bombina*, the tree frog *Hyla arborea*, the common spadefoot *Pelobates fuscus*, the common newt *Lissotriton vulgaris*, and the great crested newt *Triturus cristatus,* but due to their rarity, the common frog and moor frog were grouped together and annotated as "brown frogs". Likewise, the edible frog, pool frog and marsh frog were categorized as "green frogs" (*Rana esculenta complex*). Of all of the remaining species, the yellow-bellied toad *Bombina variegata*, alpine newt *Ichthyosaura alpestris* and carpathian newt *Lissotriton montandoni* were excluded from the analysis because they could not be easily grouped with other species and they appeared in at most 5% of the analyzed water reservoirs. Finally, we obtained seven unique species indicators which were then used for each of the 189 annotated habitats.

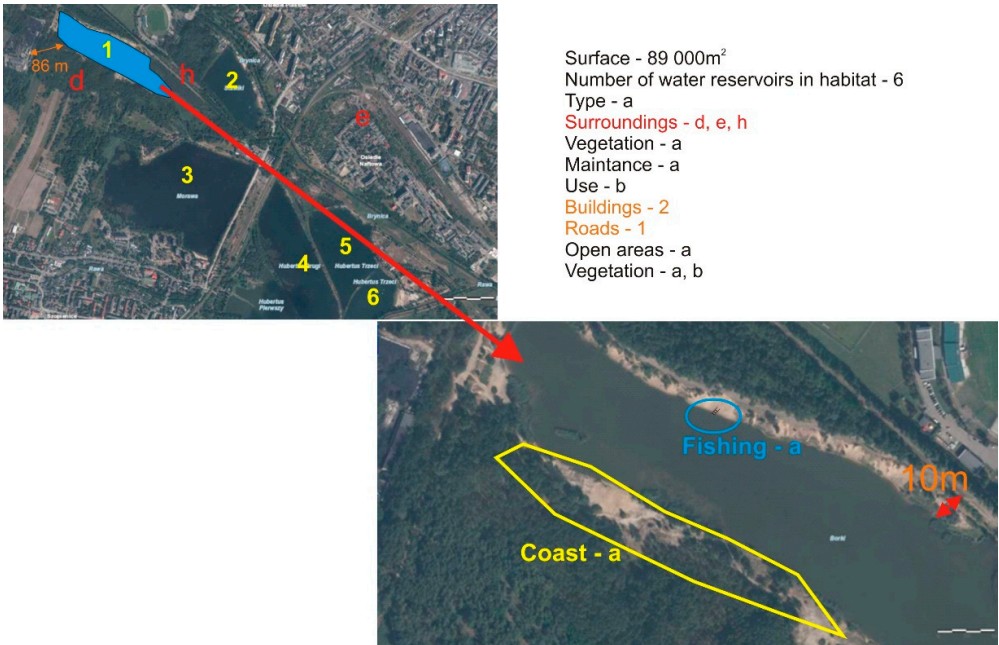

**Figure 4.** Assessment of sample water reservoirs, (adapted from [39]).

### 3.2. Methods Used in the Experiments

The experiments presented in Section 4 were divided into two separate groups. In the first group we determined the best classification method. Then, this method was used in the second experiment where we assessed the qualities of the proposed attributes. Below we provide the details of the methods and quality measures used in the experiments.

#### 3.2.1. Prediction Models

In order to build a mathematical model able to predict the presence of amphibians, we used the decision tree based machine learning models [40], including ensembles of tress. These types of models are insensitive to the attribute types and can handle numerical, ordinal and categorical features, which appear in the dataset described above. Other methods like neural networks [41], distance-based algorithms [42] or kernel-based methods (like support vector machines) [43] require an initial preprocessing stage which transforms non numerical attributes into numerical ones. Unfortunately, the preprocessing introduces extra variance to the system and can decrease the overall performance. Moreover, the decision tree based methods, such as the random forest (RF) [44] were also used by other researchers, who reported a high quality application to species distribution modelling [21]. RF

constructs a set of decision trees, which are trained independently on the data sampled from the training data in terms of features or samples. Another popular decision tree based model is boosting (e.g., AdaBoost) [45]. This ensemble adds one tree after another, so that each new tree is trained on the dataset sampled from the entire dataset, while the sampling distribution depends on the results obtained by the previously trained trees (the misclassified instances have a higher probability of being selected). More recent advances in building ensembles of the decision trees have led to so-called gradient-boosted trees (GBT) [46], which also belong to the category of boosted methods except that the trees are trained so that each new tree minimizes the gradient of an error function. This makes it even more powerful than the AdaBoost algorithm.

### 3.2.2. Performance Assessment of Prediction Models

In the experiments, we used two indicators to assess the quality of the models. These were the area under the receiver operating characteristic curve denoted as AUC and balanced accuracy (BAcc). We did not use simple accuracy, given that it is not an appropriate measure for imbalanced classification problems [47].

The advantages of applying AUC to species distribution modeling were discussed, for example, in Reference [48]. It is much more informative than the popular classification accuracy measure (a ratio between correctly classified instances and all evaluated instances) because the classification accuracy is sensitive to label distribution. The AUC evaluates how well model predictions discriminate between two classes without directly setting up a threshold, indicating a predicted class label. It is one of the most widely used threshold independent evaluators of model discriminatory power [49] and is used, in particular, in class imbalance problems. Furthermore, in the real application scenarios presented in this paper, the prediction model is intended to return the conditional probability of the final decision, rather than a simple presence/absence indicator, bearing in mind that probability is a better tool for describing the surroundings of the water reservoir and it involves human interpretation.

Balanced accuracy [47] is another commonly used measure for unbalanced classification problems. It is also known as mean recall or mean sensitivity, as well as an inverse of balanced error rate (BErr). It is obtained by calcualting mean over recalls for each class separately. The recall is the ratio between all correctly classified samples belonging to a given class and all samples originally belonging to that class. Therefore, it measures the classification performance of each class independently and calculated the average.

It must be noted, that the AUC and the BErr were previously used in the "Performance prediction challenge" [50] as reference quality measures where all the datasets were imbalanced.

### 3.2.3. Feature Quality Assessment

As reported in the literature, there are numerous methods that can be used for a feature quality assessment. A large group comprises the ranking methods [51] which assess each feature independently, taking into account some evaluation measures that estimate the relation between each individual feature and class label. An overview of the ranking methods can be found in Reference [52]. However, these methods ignore feature interdependence so they are not recommended for the purpose of our problem because important relationships between feature groups can be overlooked.

Very powerful for identifying attribute quality are feature selection wrappers, which evaluate external classifiers to determine the best feature subset [52,53]. These methods support attribute interdependence but do not return an indicator for each feature, rather a subset of attributes are assessed and ranked.

One of the advantages of the decision trees is the built in feature selection mechanism that can be transformed into a feature quality measure. The feature quality of the tree is calculated by summing up changes in risk due to splits regarding every attribute and dividing the sum by the number of branch nodes. In the case of the ensemble, it is calculated in relation to the members by averaging all quality measures returned by a single decision tree. However, the interpretation of the feature importance

returned by the tree based model can be affected by feature collinearity. The prediction model can interchangeably use one or the other of the collinear features in the tree nodes, thus redacting the feature utilization statistic. In our case, this effect was reduced by a manual feature construction, but to indicate a real relation between the features, we also calculated the mutual information (MI) matrix. Its main advantage over other measures is support for the categorical features, which in our case constituted the majority of the features. The mutual information can be calculated as follows:

$$MI(X,Y) = \sum_{x,y} \frac{P_{XY}(x,y)log(P_{XY}(x,y))}{P_X(x)P_Y(y)} = H(X) - H(X|Y)$$

where $P_{XY}(x,y)$ is the joint probability distribution between variables $X$ and $Y$. Alternatively $H(X)$ is the Shanon's entropy of variable $X$, and $H(X|Y)$ is the conditional entropy of $X$ knowing $Y$.

Values of this type of matrix represent mutual information between any feature $i$ and feature $j$ MI $(f_i, f_j)$, that is, the reduction of uncertainty of one variable knowing the other, expressed in terms of bits.

### 3.2.4. Experiment Design

The experiments were divided into two stages. In the first stage, we assessed the quality of the four classification methods described in Section 3.2.1, the decision tree, AdaBoost, RF and GBT, in relation to the dataset described in Section 3.1. The second stage was concerned with rating the input features in order to determine which attributes were the most important.

To assess the quality of the prediction models, we used the process presented in Figure 5, which starts by loading the data, after which the model in question is trained and tested using the k-fold cross-validation procedure. This procedure is based on splitting the dataset into k disjoint random subsets. Then, in k iterations, the model is trained on k-1 subsets and its performance is evaluated on a remaining subset. After k iterations are obtained, the performances are averaged and reported. As all of the evaluated models needed hyper-parameter selection and optimization to avoid overfitting, the optimization procedure was embedded into the main cross-validation procedure. For each iteration the model parameters were optimized using a grid search algorithm, using the internal optimization process to achieve the maximum AUC. To speed up the optimization process, the internal cross-validation had k = 5 folds. As indicated above, we evaluated the C4.5 decision tree for which the confidence was optimized in the range 0.001 to 0.45 [40], for the RF the size of the ensemble (between five and 40) and the number of considered features were optimized, for Ada-Boost we evaluated the size of the ensemble and the confidence of the C4.5 decision tree and, finally, for the GBT we optimized the size of the ensemble, learning rate and the tree confidence. For the GBT, the balance dataset option was turned on. The feature quality factors were obtained with the best prediction model for a given species.

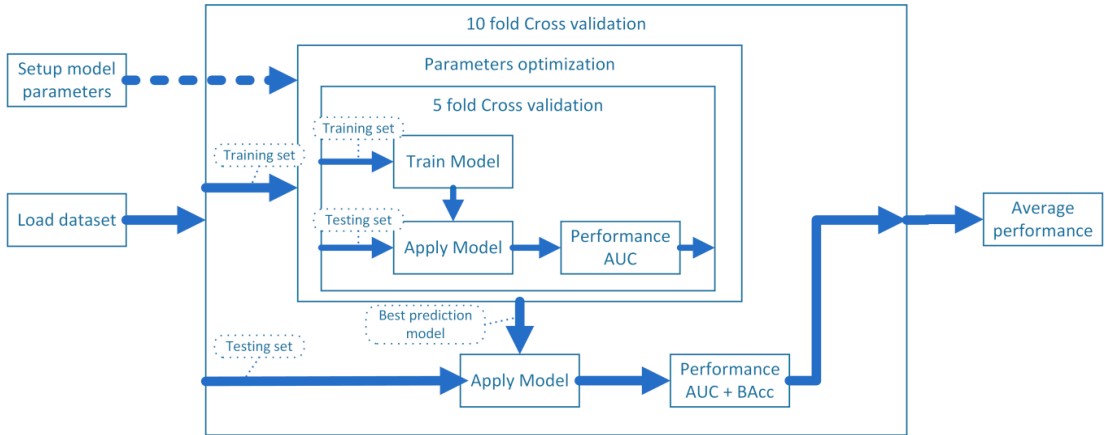

**Figure 5.** A process of prediction models quality assessment.

The experiments were conducted using the RapidMiner software (https://rapidminer.com/) as a shell system, with a Weka plug-in. All the models originated from Weka, except GBT, which was taken from the H$_2$O project (the H$_2$O project is built into the RapidMiner software).

## 4. Results and Discussion

The results obtained for model comparison are presented in Table 2, where the best model for each of the species is marked in bold. As can be seen in terms of the AUC, GBT usually lead; only for the common newt, the green frog and the great crested newt did the random forest outperform GBT. In terms of balanced accuracy GBT also lead, except the earlier mentioned species for which, respectively, random forest, AdaBoost and single decision tree overtook GBT. In all of the cases where GBT did not lead, the difference was not statistically significant (we used the Welch's t-test with $\alpha = 0.1$). Only for the green frog, in terms of the AUC, did RF significantly outperform GBT.

**Table 2.** The performances obtained for four evaluated classification models for each of the amphibian species. The values represent area under curve (AUC).

| | AUC | | | | Balanced Accuracy | | | |
|---|---|---|---|---|---|---|---|---|
| | **GBT** | **RF** | **ADA** | **DT** | **GBT** | **RF** | **ADA** | **DT** |
| Green frogs | 68.80% | **75.92%** | 68.28% | 62.13% | 66.51% | **67.44%** | 66.36% | 62.65% |
| Brown frogs | **63.63%** | 56.04% | 58.79% | 48.55% | **60.58%** | 54.61% | 54.56% | 52.46% |
| Common toad | **71.56%** | 63.50% | 68.80% | 67.78% | **64.57%** | 60.73% | 62.08% | 62.23% |
| Fire-bellied toad | **65.71%** | 58.77% | 56.79% | 53.18% | **68.34%** | 52.99% | 54.93% | 56.64% |
| Tree frog | **67.17%** | 63.07% | 61.94% | 57.53% | **60.24%** | 57.43% | 55.32% | 59.82% |
| Common newt | 64.90% | **66.35%** | 61.06% | 62.84% | 61.44% | 54.80% | 58.66% | **62.97%** |
| Great crested newt | 83.10% | **86.97%** | 77.47% | 51.00% | 67.56% | 54.76% | **68.15%** | 51.79% |

These results allowed for a selection of the GBT classifiers to be used in the second stage of experiments. As indicated in Section 3.2.3, we started the second stage by calculating the mutual information matrix (see Table 3). The highest MI values were obtained between distances to roads (RR) and buildings (BR) and between Fishing (FR) and Use (UR). For these pairs, the MI was around 0.65, which is relatively significant, considering that the entropy of UR is 1.1 and the entropy of FR is 1.6. For the RR and BR we also calculated the Pearson's correlation coefficient, which was equal to 0.69. This value can be explained by the fact that buildings usually occur near roads.

**Table 3.** The mutual information matrix representing the mutual information (MI) values between each pair of attributes of the evaluated dataset.

| | SR | NR | OR | RR | BR | TR | VR | SUR1 | SUR2 | SUR3 | UR | FR | MR | CR |
|---|---|---|---|---|---|---|---|---|---|---|---|---|---|---|
| SR | - | 0.130 | 0.025 | 0.028 | 0.029 | 0.021 | 0.049 | 0.057 | 0.050 | 0.064 | 0.034 | 0.071 | 0.001 | 0.001 |
| NR | 0.130 | - | 0.064 | 0.105 | 0.074 | 0.096 | 0.116 | 0.122 | 0.134 | 0.194 | 0.113 | 0.216 | 0.005 | 0.003 |
| OR | 0.025 | 0.064 | - | 0.195 | 0.150 | 0.098 | 0.050 | 0.366 | 0.158 | 0.165 | 0.028 | 0.066 | 0.028 | 0.022 |
| RR | 0.028 | 0.105 | 0.195 | - | 0.669 | 0.191 | 0.111 | 0.236 | 0.167 | 0.162 | 0.042 | 0.051 | 0.014 | 0.007 |
| BR | 0.029 | 0.074 | 0.150 | 0.669 | - | 0.225 | 0.117 | 0.226 | 0.145 | 0.150 | 0.100 | 0.082 | 0.017 | 0.015 |
| TR | 0.021 | 0.096 | 0.098 | 0.191 | 0.225 | - | 0.390 | 0.329 | 0.228 | 0.152 | 0.270 | 0.254 | 0.015 | 0.032 |
| VR | 0.049 | 0.116 | 0.050 | 0.111 | 0.117 | 0.390 | - | 0.125 | 0.178 | 0.191 | 0.286 | 0.316 | 0.035 | 0.026 |
| SUR1 | 0.057 | 0.122 | 0.366 | 0.236 | 0.226 | 0.329 | 0.125 | - | 0.279 | 0.271 | 0.044 | 0.088 | 0.020 | 0.018 |
| SUR2 | 0.050 | 0.134 | 0.158 | 0.167 | 0.145 | 0.228 | 0.178 | 0.279 | - | 0.281 | 0.081 | 0.145 | 0.024 | 0.022 |
| SUR3 | 0.064 | 0.194 | 0.165 | 0.162 | 0.150 | 0.152 | 0.191 | 0.271 | 0.281 | - | 0.053 | 0.157 | 0.057 | 0.045 |
| UR | 0.034 | 0.113 | 0.028 | 0.042 | 0.100 | 0.270 | 0.286 | 0.044 | 0.081 | 0.053 | - | 0.638 | 0.015 | 0.011 |
| FR | 0.071 | 0.216 | 0.066 | 0.051 | 0.082 | 0.254 | 0.316 | 0.088 | 0.145 | 0.157 | 0.638 | - | 0.015 | 0.016 |
| MR | 0.001 | 0.005 | 0.028 | 0.014 | 0.017 | 0.015 | 0.035 | 0.020 | 0.024 | 0.057 | 0.015 | 0.015 | - | 0.033 |
| CR | 0.001 | 0.003 | 0.022 | 0.007 | 0.015 | 0.032 | 0.026 | 0.018 | 0.022 | 0.045 | 0.011 | 0.016 | 0.033 | - |

Finally, in order to assess the quality of the attributes, we used the GBT ensemble, and trained it again using the entire dataset (without cross-validation), separately for each label attribute. Out of each obtained model, the feature importance indicators were recorded and the results are presented in Table 4. Note that the feature importance factors were normalized to fit the range [0,1] for each species in order to make them comparable across different species.

**Table 4.** Feature importance indicators for each of the amphibian species.

| Feature | Green frogs | Brown frogs | Common toad | Fire-bellied toad | Tree frog | Common newt | Great Crested newt | Average |
|---|---|---|---|---|---|---|---|---|
| SR | 0.90 | 0.34 | 0.09 | 1.00 | 0.00 | 0.34 | 0.74 | 0.49 |
| NR | 0.00 | 0.08 | 0.02 | 0.07 | 0.00 | 0.17 | 0.58 | 0.13 |
| TR | 1.00 | 0.93 | 1.00 | 0.46 | 0.58 | 1.00 | 0.49 | 0.78 |
| VR | 0.31 | 0.78 | 0.59 | 0.47 | 0.50 | 0.65 | 0.55 | 0.55 |
| **SUR 1** | **0.52** | **0.55** | **0.87** | **0.59** | **1.00** | **0.76** | **1.00** | **0.76** |
| **SUR 2** | **0.61** | **0.96** | **0.81** | **0.63** | **0.54** | **0.84** | **0.76** | **0.74** |
| **SUR 3** | **0.61** | **1.00** | **0.57** | **0.64** | **0.19** | **0.61** | **0.47** | **0.58** |
| UR | 0.11 | 0.25 | 0.01 | 0.02 | 0.00 | 0.11 | 0.01 | 0.07 |
| FR | 0.39 | 0.03 | 0.44 | 0.25 | 0.00 | 0.47 | 0.23 | 0.26 |
| **OR** | **0.10** | **0.18** | **0.00** | **0.24** | **0.00** | **0.54** | **0.16** | **0.18** |
| RR | 0.29 | 0.23 | 0.34 | 0.24 | 0.17 | 0.37 | 0.37 | 0.29 |
| BR | 0.11 | 0.12 | 0.05 | 0.12 | 0.00 | 0.28 | 0.47 | 0.16 |
| **MR** | **0.00** | **0.00** | **0.00** | **0.00** | **0.00** | **0.00** | **0.00** | **0.00** |
| **CR** | **0.00** | **0.00** | **0.00** | **0.00** | **0.00** | **0.00** | **0.00** | **0.00** |

Then, these results were averaged over all amphibian species, such that the averaged value represented the final feature importance (the last column in the table). Next, the feature importance indicators were ranked from most to least important and visualized in Figure 6.

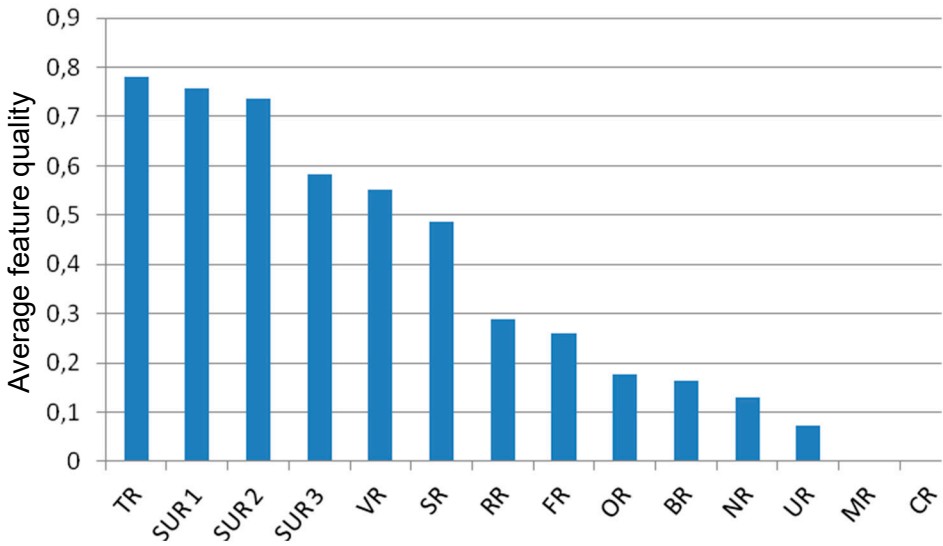

**Figure 6.** Final feature importance.

The obtained values indicate that the most important features are: type of water reservoir (TR) and its surroundings—SUR 1, SUR 2 and SUR 3—followed by vegetation (VR) and surface. It is important to note that three types of surroundings, which were introduced for the first time, are among the top features. The features of average importance include: the occurrence of buildings, roads, forests, wasteland, and meadows. Interestingly they are relatively simple to obtain from satellite images and make a composite of the most important factors. Moreover they have much more

discriminative power than the distance to buildings (BR) or roads (RR) because it is unknown how many building/roads surround the habitat. Such results were expected because of the amphibian requirements of the habitat [54–59]. The type of reservoir reflects the quality of the habitat, while the surroundings characterize the possibility of access to the breeding site as well as the quality of the land habitat in the vicinity of the reservoir. Favorable as well as unfavorable conditions in both of these habitats have a direct impact on the occurrence of amphibians. From the herpetologist point of view, it was expected that the importance of distance from a water reservoir to the buildings would be more significant [4], but the two variables BR and RR are strongly related (as mentioned above the Pearson's correlation coefficient was 0.69) so the obtained importance indicators may be underestimated.

It should be noted that the obtained results were averaged over all considered species, but some of them may be valid only for certain species. A perfect example to illustrate this thesis is the variable surface (SR), which for many species like green frog, fire-bellied toad or great crested newt is among the most important factors. These results can be explained by the fact that the fire-bellied toad and the pool frog (Pelophylax lessonae), which belong to the green frog group, prefer small water reservoirs, and the great crested newt prefers larger ponds [54,60]. Our results suggest that for common toads and tree frogs the surface is not an important indicator. This is in contrast with common knowledge because both species prefer larger water reservoirs [54,60]. However, it applies to lowland areas and in sub-mountainous areas (which were included in the evaluation) these species also use small reservoirs to reproduce [54,60].

The second variable with the highest variance was the type of water reservoir (TR), which in general was the most important variable. It influenced all species, so its importance often reached 1, but for the fire-bellied toad and the great crested newt its importance was below 0.5. Another feature of that type is the open area (OR), with an average importance of 0.16. If we pay attention to the values of the individual species, it turns out that this feature plays an important role in discriminating between the presence and absence of the common newt. In general this feature is considered very important from the human expert point of view [4]. However, similar information can be induced from the combination of SUR1-3 variables (this relation appears between the single variable OR and combined SUR1-3 so it is not expressed in the mutual information matrix).

Vegetation (marked as VR) is an important feature of a water habitat of amphibians. It refers to such species as the great crested newt, the tree frog, the fire-bellied toad and the brown frog (the common frog and the moor frog) [54,60]. For a pond to be attractive to amphibians, the overgrown fragments of the water table will suffice. We obtained values of the VR parameter at the level of 0.5–0.6, which can be considered a satisfactory value. The highest VR value was 0.78 and refers to brown frogs that spawn in the overgrown parts of the pond [54,58,59]. We received the lowest VR value for green frogs that accept reservoirs that are poorly overgrown or even devoid of vegetation.

On the basis of the calculations, two features, indicated by experts, proved to be significant: the maintenance of the water reservoir (MR) and the type of coast (CR). These results can be confusing but, due to the fact that most of the analyzed reservoirs were not littered, the state of maintenance was determined to be good so the algorithm was not able to discover this dependence. The shores of the water reservoirs were also mostly natural. See Figure 7, which shows the distribution of the categorical features. In the case of the type of coast (CR), amphibians were found in reservoirs made with concrete and along natural coastlines. This variable may be important for the population size assessment, but does not offer any advantages when assessing the presence or absence of species.

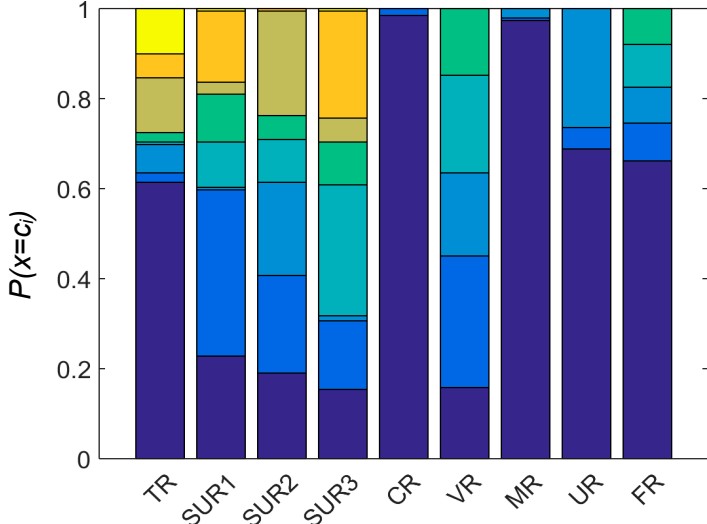

**Figure 7.** The probability distribution $P(x = c_i)$ of categorical variables. Each bar represents a single variable, the number of colors represents the unique value $c_i$ of a given feature, and the height of each color bar represents the frequency of value $c_i$. For details see the description presented in Table 1.

## 5. Conclusions

Amphibians, as some of the most vulnerable animal phyla, and which are protected by law, are sensitive to the transformation of the environment. The presence of numerous species of amphibians in water reservoirs indicates the presence of other valuable species. To effectively prepare a natural inventory, it is important to scrupulously carry out field studies.

The presence of amphibians is a feature that can be the basis of any natural evaluation. As a rule, the greater the number of amphibian species and populations being analyzed, the more valuable the habitat is.

Common problems include assessing the presence of amphibians in the area of planned investments and verifying the results of inventories. In order to address such issues, we have proposed a methodology that utilizes publicly available data from GIS and satellite images to estimate the presence or absence of selected species of amphibians. For this purpose, we have suggested a feature space, which covers the most significant factors affecting the habitats. Species distribution modeling has become a useful method for predicting amphibian ranges, based on the relationships between species records and environmental variables. A selection of features that we (based on the recommendation of experts) and other scientists considered relevant was taken into account. Our results indicate that the selected feature set has enough discrimination power to be used for real-life applications. Moreover, as it is presented and discussed in Section 4, the estimated relevance of particular features is justified by expert knowledge and reflects the preferences of amphibians. Only two out of all of the features evaluated turned out to be irrelevant, but this may be the result of an unfavorable distribution of these features in the analyzed dataset. It requires further investigation with extended dataset. The analysis indicated that the gradient-boosted trees were the best method to discriminate the presence or absence of amphibians in the habitats. The mean value of the AUC for the analyzed dataset for this method was equal to 0.693, which is comparable to the results obtained by other authors (for example, see [21,55]), but in our case the input features were gathered remotely.

In summary, we hope that the proposed solution will help improve the quality of the inventory work conducted before any infrastructure project. We also hope state institutions will utilize these tools for initial screening tests to assess the validity of the EIA. It should be stipulated that the assessment of the habitat requires experience in the field of herpetology and the results obtained by the proposed method should be subjected to critical analysis.

**Author Contributions:** Conceptualization—Marek Sołtysiak, Data analysis—Marcin Blachnik, Marek Sołtysiak, Dominika Dąbrowska, Investigation—Marek Sołtysiak, Methodology—Marcin Blachnik, Marek Sołtysiak, Dominika Dąbrowska, Software—Marcin Blachnik.

**Funding:** This research received no external funding.

**Acknowledgments:** The research was supported by the Centre for Polar Studies, University of Silesia, Poland. The Leading National Research Centre (KNOW) in Earth Sciences 2014–2018.

**Conflicts of Interest:** 'The authors declare no conflicts of interest.

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
