# Peer review of "Predicting Presence of Amphibian Species Using Features Obtained from GIS and Satellite Images"

_ijgi, doi:10.3390/ijgi8030123_

Round 1

Reviewer 1 Report

The manuscript entitled "Predicting presence of amphibian species using features obtained from GIS and satellite images" attempts to show a use case of machine learning methods to predict presence of amphibian species while obtaining features through manual digitizing over online satellite images services. The authors claim that this approach would enhance EIA studies and they present some new variables that have not been used before (so they say).

While the topic seems interesting and important for amphibians' conservation in the light of roads projects and enhancing some EIA studies, the manuscript is very poorly written, there's no clearly stated objective, terms used are not defined properly (e.g., habitat, open areas), variable definitions are vague (e.g., table 1), figures, esp. maps, are very noisy and impossible to read, methods are not clearly explained (do the authors estimate all distances and areas manually?), there are parts of the MDPI template that the authors didn't remove and, there's no discussion at all (hence, no comparison with other studies, no reflection about limitations of the approach, etc.). Given all these reasons, I recommend rejection. See further corrections, comments and questions below.

Introduction:

l33: to manage "an area" (to manage a space sounds odd)

l38: replace "constitute areas" by "be more suitable"

l39-43: fix these 2 sentences, they mix topics and are too long. Please split them properly.

l46: Which requirements? You do not mention any requirement before. Also, do not start a new paragraph using "these", if a new paragraph start a new topic or make it explicit what you are referring to.

l58-67: when mentioning difficulties, make examples more explicit for those not knowing about amphibians ecology, e.g., "after a long and cold winter abundances are expected to be low/very low"

l71: fix sentence, you do not provide the methods but an example of application of established methods

l73: remove extra comma

l73: remove The, capitalize most

l74: satellite "imagery", remove using, put the website in parenthesis. What are "urban spatial development plans"? Any link or example?

l75-77: split that sentence, too much information

l77: replace appearance with "occurrence"

l80: No model was mentioned earlier. Which model are you talking about?

l81: ... other EIA procedures. Such as? Add examples.

l82: what is linear investment planning?

l90-91: AFAIU, you obtain variables through visual analysis and digitizing/measuring over very high resolution images available online. Which images are those btw?

l94: Unless this dataset is made available, I do not think this is a relevant discussion topic.

l95-100: How the manuscript is organized is not relevant. Remove this paragraph and better state your objective instead.

l106: remove "the"

l108: what do you mean by "small range"? Set a scale in parenthesis

l111: ... or using the maximum entropy "algorithm to predict."

l114-120: annual mean precipitation is both in Type I and Type II groups

l118: replace who by which (who is used only for persons)

l125: Type II (not I)

l126: What do you mean with "included in a GIS framework"? GIS software do not come with variables included, one uses their tools over data (raster or vector) to obtain variables relevant to one's scope.

l129: high volatility??

l131: not all sensors require constant control of readings. More and more sensors are nowadays set to send automatic reports.

l139-151: Please remove this; it's part of the journal template.

Material and methods:

l155: similar comment as for l126.

l156: same as before; what are the "spatial development plans"?

l164: fix verbs tenses, i.e., remove s

l165: Then "these" features

l173-178: provide more details, source of data, links, etc.

l177: "amphibian" (?) population

l183: ... and "it is" about 75 km long

Figure 2 and 3: these are very noisy, they look like pictures of a folded paper map. There are nowadays very nice base maps and GIS software (even online options) that can be used to produce proper maps. Please fix, as they are, they are not suitable for publication.

l198: please fix the sentence

l209-220: Variable names requires enhancement and more precision, e.g., I would change "Three dominant types of surroundings of water reservoirs" to " first|second|third dominant type of land cover in the surrounding of the water reservoir"; "Vegetation in the reservoirs" to "presence of vegetation in the water reservoir"; "State of Maintenance of the reservoir" to "Maintenance status" and so on...

l222: Categorical is usually used, not symbolic

Table 1:

- What concept of habitat are you using? It seems to me you are mostly referring to the study area (instead of habitat) when you count the number of water reservoirs.

- Please fix verbs and enhance description of "Type" variable

- How can they "see" the maintenance status of reservoirs, the different types of vegetation within the reservoir, the type of use, or the presence of fishing?? I've checked the site they provide and it's not better that bing, esri or google earth imagery. This is really not clear and should be explained or discussed properly.

- Do you measure distance by hand? Do you use the minimum distance or any distance?

- What do you mean by open areas and how do you quantify the access percentage from the edges? What does 0% means; all the borders have no access to open areas? What does access mean in this context?

l238: Describe where you obtained the amphibians data, provide a link or source. How was that data collected, i.e., samplings, recordings??

Figure 4: Vegetation is characterized as both a and b...

l314-316: again... remaining of journal template

Results:

l318-330: These 2 paragraphs correspond to methods, not results. Provide references for software used.

Figure 5: Please fix caption, it is not clear. Moreover, where do the arrows coming out from "performance measure" go?

Figure 6: not needed, the information is already in table 4.

Figure 7: What does the y axis represent here?

Prediction maps should be presented so that the reader gets an idea of where each species is more likely to be present.

l366: average "importance"

Discussion/Conclusion:

Some discussion is included in the results section, but it lacks depth and proper referencing. Indeed, there are no references at all (neither in the Conclusion). There's no comparison with other studies and no acknowledgement of limitations of the proposed approach. Also, discussion about differences among species would be desirable, especially for those endangered species, if any.

Author Response

Introduction:

l33: to manage "an area" (to manage a space sounds odd)

This sentence has been changed.

l38: replace "constitute areas" by "be more suitable"

It was replaced.

l39-43: fix these 2 sentences, they mix topics and are too long. Please split them properly.

These sentences have been changed.

l46: Which requirements? You do not mention any requirement before. Also, do not start a new paragraph using "these", if a new paragraph start a new topic or make it explicit what you are referring to.

It has been removed.

l58-67: when mentioning difficulties, make examples more explicit for those not knowing about amphibians ecology, e.g., "after a long and cold winter abundances are expected to be low/very low"

That part has been corrected. Several aspects of amphibians’ biology have been described.

l71: fix sentence, you do not provide the methods but an example of application of established methods

It has been changed.

l73: remove extra comma

It was removed.

l73: remove The, capitalize most

It was removed.

l74: satellite "imagery", remove using, put the website in parenthesis. What are "urban spatial development plans"? Any link or example?

That part has been corrected and extra link has been added.

l75-77: split that sentence, too much information

It has been changed.

l77: replace appearance with "occurrence"

It has been replaced.

l80: No model was mentioned earlier. Which model are you talking about?

It has been explained.

l81: ... other EIA procedures. Such as? Add examples.

It has been removed.

l82: what is linear investment planning?

This sentence has been rearranged.

l90-91: AFAIU, you obtain variables through visual analysis and digitizing/measuring over very high resolution images available online. Which images are those btw?

It has been corrected.

l94: Unless this dataset is made available, I do not think this is a relevant discussion topic.

It has been removed but this dataset will be available.

l95-100: How the manuscript is organized is not relevant. Remove this paragraph and better state your objective instead.

That part has been changed.

l106: remove "the"

It has been removed.

l108: what do you mean by "small range"? Set a scale in parenthesis

That part has been changed.

l111: ... or using the maximum entropy "algorithm to predict."

It has been added.

l114-120: annual mean precipitation is both in Type I and Type II groups

It has been corrected.

l118: replace who by which (who is used only for persons)

It has been replaced.

l125: Type II (not I)

It has been corrected.

l126: What do you mean with "included in a GIS framework"? GIS software do not come with variables included, one uses their tools over data (raster or vector) to obtain variables relevant to one's scope.

It has been corrected.

l129: high volatility??

It has been corrected.

l131: not all sensors require constant control of readings. More and more sensors are nowadays set to send automatic reports.

It has been corrected.

l139-151: Please remove this; it's part of the journal template.

It has been removed.

Material and methods:

l155: similar comment as for l126.

It has been corrected.

l156: same as before; what are the "spatial development plans"?

It has been corrected.

l164: fix verbs tenses, i.e., remove s

It has been removed.

l165: Then "these" features

It has been corrected.

l173-178: provide more details, source of data, links, etc.

More details have been added.

l177: "amphibian" (?) population

It has been added.

l183: ... and "it is" about 75 km long

It has been changed.

Figure 2 and 3: these are very noisy, they look like pictures of a folded paper map. There are nowadays very nice base maps and GIS software (even online options) that can be used to produce proper maps. Please fix, as they are, they are not suitable for publication.

These figures have been changed.

l198: please fix the sentence

It has been changed.

l209-220: Variable names requires enhancement and more precision, e.g., I would change "Three dominant types of surroundings of water reservoirs" to " first|second|third dominant type of land cover in the surrounding of the water reservoir"; "Vegetation in the reservoirs" to "presence of vegetation in the water reservoir"; "State of Maintenance of the reservoir" to "Maintenance status" and so on...

Names have been changed, and more detailed description is provided

l222: Categorical is usually used, not symbolic

It has been changed.

Table 1:

- What concept of habitat are you using? It seems to me you are mostly referring to the study area (instead of habitat) when you count the number of water reservoirs.

An Amphibians population demonstrate attachment to a specific water reservoir - amphibians return to their birthplace to breed (philopatry). The water reservoir is an essential element of the habitat, however, its necessary complement is a land habitat. For this reason, the authors analyzed the features of a set of many habitats of amphibians located in the strip of land along the road investment.

- Please fix verbs and enhance description of "Type" variable

It has been changed.

- How can they "see" the maintenance status of reservoirs, the different types of vegetation within the reservoir, the type of use, or the presence of fishing?? I've checked the site they provide and it's not better that bing, esri or google earth imagery. This is really not clear and should be explained or discussed properly.

We have corrected the content in the table. We should bear in mind that when analyzing satellite images, we should supplement our information with results from field inspection. so we wrote that field inspection is very important for correct assessment of habitat features. Satellite images do not contain all information, such as the presence of fish, fishing activity, or rubbish and they may be out of date additionatelly (L:266-268).

- Do you measure distance by hand? Do you use the minimum distance or any distance?

We proposed using minimum distance ranges - their range is so wide that it allows taking into account the variability of the object's distance from the collective. We additionally wrote, that „Geodetic services and GIS applications have tools to determine parameters such as surface or distance. It is a good practice to use several sources of information - satellite maps  presented there may come from different periods and may show the variability of the analyzed features of the habitat. Additionally, we wrote that features of amphibians habitats can be determined by a satellite map analysis and during a single inspection of the site.

- What do you mean by open areas and how do you quantify the access percentage from the edges? What does 0% means; all the borders have no access to open areas? What does access mean in this context?

We changed the name of the feature to „access from water table to land habitats”. 0% means that there is no access to the reservoir – e.g. it is surrounded by buildings.

All remarks connected with Table 1 have been taken into account.

l238: Describe where you obtained the amphibians data, provide a link or source. How was that data collected, i.e., samplings, recordings??

This information has been added to the text. We are on the process to publishing the data to the UCI repository. https://archive.ics.uci.edu/ml/index.php

Figure 4: Vegetation is characterized as both a and b...

A more detailed description was added to the text.

l314-316: again... remaining of journal template

It has been removed.

Results:

l318-330: These 2 paragraphs correspond to methods, not results. Provide references for software used.

The references have been added, and the paragraphs are rearranged

Figure 5: Please fix caption, it is not clear. Moreover, where do the arrows coming out from "performance measure" go?

Figure 5 is replaced with a new one with more detail description. The paragraph describing the performance evaluation is rewritten The performance comparison calculations are re-evaluated according to the corrected process. In the corrected process the hyper-parameter optimization process is embedded into the cross validation procedure to avoid overestimation of the results.

Figure 6: not needed, the information is already in table 4.

We agree that figure 6 is redundant and its values are presented in table 5 (we added one more table – the mutual information matrix), but our aim is to visualize the differences between the importance of particualr features. It also order the features from the most to the least important.

Figure 7: What does the y axis represent here?

The axis description have been added to the figure

l366: average "importance"

It has been changed.

Discussion/Conclusion:

Some discussion is included in the results section, but it lacks depth and proper referencing. Indeed, there are no references at all (neither in the Conclusion). There's no comparison with other studies and no acknowledgement of limitations of the proposed approach. Also, discussion about differences among species would be desirable, especially for those endangered species, if any.

The obtained results were referred to selected species of amphibians. Authors focused on the application of habitat assessment methods, not on the ecology of amphibians. We are aware that the collected data is mainly of qualitative data and that further field research is necessary. In practice, however, for the needs of the EIA report, it is assumed that the qualitative data is sufficient. It should be remembered that amphibians are endangered as a group of animals. One of the main causes of extinction of amphibians is the destruction of habitats.

Reviewer 2 Report

This paper tests several different machine learning approaches to predict the presence of amphibian species using a set of environmental predictors.

Major comments:

--- Grammar and spelling mistakes can be seen throughout the manuscript-- at times, it distracts from the content. For instance, in line 42 “the most important being the transformation of habitats”, in line 118 the word “who”, or lines 140-151 must be deleted, etc. Scientific proofreading and editing by a native speaker are necessary.

--There are numerous statements without references across the manuscript. As an example, from Lines 30-38 most sentences require references.

--How authors can make sure that overfitting has not occurred in the developed models?

--It is not clear if data division is only performed on your training + validation data, not your test data. Clarify, please.

-- It seems that the authors haven’t tested for the independency of factors before running their models which would end up with multicollinearity and redundancy which undermines the significance of independent variables.

-- How did you fine-tune model’s hyperparameters?

--Which one of the findings is actionable? How do you take it to the next step in terms of ecological implications?

---AUC has been criticized in literature when the number of samples between categories varies significantly, I would suggest incorporating some other evaluation metrics. For binary classification models in addition to AUC, accuracy matrix and its derivatives (sensitivity, specificity, overall accuracy, kappa statistics) could be evaluated and applied.

Minor comments:

--The term “feature obtained from GIS” mistakenly implies that GIS makes environmental features. I think it should be replaced with other appropriate terms such as “features generated in GIS environment”.

--Line 14. ”and other issues”: Authors should avoid using the ambiguous terms, remove it or mention the issues.

--Lines 73-76 should be moved to the Methods section.

–Borders of 500m and 1000 meters in the legend of Fig 2 are not distinguishable on the map

–Lines 98 to 100 are redundant and should be removed.

--Line 125—What do you mean by “Land use—Type I”?

--In line 84 “Figure 1” should be changed to “Figure 2”.

Author Response

--- Grammar and spelling mistakes can be seen throughout the manuscript-- at times, it distracts from the content. For instance, in line 42 “the most important being the transformation of habitats”, in line 118 the word “who”, or lines 140-151 must be deleted, etc. Scientific proofreading and editing by a native speaker are necessary.

Grammar and spelling mistakes have been corrected.

--There are numerous statements without references across the manuscript. As an example, from Lines 30-38 most sentences require references.

References have been added.

--How authors can make sure that overfitting has not occurred in the developed models?

The experimental process is changed. Now, the hyper-parameter optimization procedure of the prediction model is embedded into the training part of the cross-validation procedure to avoid overestimation of the results. According to the best of our knowledge this scheme is the state of art in proper performance estimation.  The change of the experimental setup resulted in changes in the model comparison stage of the experiments.

--It is not clear if data division is only performed on your training + validation data, not your test data. Clarify, please.

A new description of the experiments is added. In the experiments we used two level hierarchy of the cross-validation (CV) procedure. We used the 10 fold CV procedure (outer) test to estimate the performance and we also used the internal 5 fold CV (embedded into the outer CV ) procedure to estimate the performance during the hyper-parameters optimization procedure. In this scenario the outer (10-fold cross-validation) can be interpreted as the 10 times repeated split into training/test set, and the hyper-parameter optimization procedure used internal 5-fold CV. The internal CV can be interpreted as 5 times repeated split of the training set obtained from the outer CV into training / validation set. As stated above, according to the best of our knowledge this procedure allows to avoid overfitting and provides the most accurate estimation of the performance.

-- It seems that the authors haven’t tested for the independency of factors before running their models which would end up with multicollinearity and redundancy which undermines the significance of independent variables.

The experiments are extended, and the mutual information matrix (MI) is added to the results  description. The MI matrix indicates that only two pairs of features are related. This observation is included in the human-expert  interpretation of the feature importance indicators

-- How did you fine-tune model’s hyperparameters?

Hyperparameters were optimized using grid search. The optimization procedure is embedded into the training part of the outer CV  process.

--Which one of the findings is actionable? How do you take it to the next step in terms of ecological implications?

We added a description to the text

---AUC has been criticized in literature when the number of samples between categories varies significantly, I would suggest incorporating some other evaluation metrics. For binary classification models in addition to AUC, accuracy matrix and its derivatives (sensitivity, specificity, overall accuracy, kappa statistics) could be evaluated and applied.

According to the best of our knowledge the standard accuracy is misleading in terms of imbalanced datasets, but we extended the results  providing  balanced accuracy (BAcc, also known as 1 – balanced error rate) next to AUC. BAcc is the standard performance measure for imbalanced datasets when class importance factors are unknown or equal. The balanced accuracy is also known as Mean Recall or Mean Sensitivity (average over sensitivities calculated for each class independently)

Minor comments:

--The term “feature obtained from GIS” mistakenly implies that GIS makes environmental features. I think it should be replaced with other appropriate terms such as “features generated in GIS environment”.

It has been corrected.

--Line 14. ”and other issues”: Authors should avoid using the ambiguous terms, remove it or mention the issues.

It has been removed.

--Lines 73-76 should be moved to the Methods section.

It has been changed.

–Borders of 500m and 1000 meters in the legend of Fig 2 are not distinguishable on the map

Figure 2 has been changed.

–Lines 98 to 100 are redundant and should be removed.

It has been removed.

--Line 125—What do you mean by “Land use—Type I”?

This information has been explained in the text.

--In line 84 “Figure 1” should be changed to “Figure 2”.

It has been changed.

Round 2

Reviewer 1 Report

The paper has been enhanced, but still there's no proper discussion with references and all. Readers must blindly believe what authors say, e.g., that frog X prefers this or that kind of pond. I'm fine with them writing results and discussion altogether, but then it needs to be a proper integrated discussion. And that involves comparison with other studies and assessment of limitations of their own approach in the light of existent literature.

Author Response

Dear Reviewer

Thank you very much for your comments.

We added appropriate references to the discussion part and also extended the discussion. An extra paragraph (lines 472-479) is added to the manuscript which discuss the vegetation variable. Some minor, editorial changes were also made

Reviewer 2 Report

The authors addressed my comments.

Author Response

Dear Reviewer

Thank you very much for your efforts and time spend to improve the manuscript.